# Unveiling the Relevance of the Oral Cavity as a *Staphylococcus aureus* Colonization Site and Potential Source of Antimicrobial Resistance

**DOI:** 10.3390/pathogens12060765

**Published:** 2023-05-26

**Authors:** Joana Campos, Mariana Faria Pires, Marta Sousa, Carla Campos, Carolina Fernandes Ferreira Alves da Costa, Benedita Sampaio-Maia

**Affiliations:** 1INEB—Instituto Nacional de Engenharia Biomédica, 4200-135 Porto, Portugal; joanaaraujocampos@gmail.com (J.C.);; 2i3S—Instituto de Investigação e Inovação em Saúde, Universidade do Porto, 4200-135 Porto, Portugal; 3Faculdade de Medicina Dentária, Universidade do Porto, 4200-393 Porto, Portugal; marianascfpires@gmail.com (M.F.P.); marta.dan.sousa@gmail.com (M.S.); 4Instituto Português de Oncologia do Porto Francisco Gentil, 4200-072 Porto, Portugal; carla.campos@ipoporto.min-saude.pt; 5Escola Superior de Saúde, Instituto Politécnico do Porto, 4200-072 Porto, Portugal; 6Instituto de Ciências Biomédicas Abel Salazar, Universidade do Porto, 4050-313 Porto, Portugal

**Keywords:** *Staphylococcus aureus*, oro-nasal carriage, antibiotic resistance, multidrug-resistance, healthy individuals

## Abstract

*Staphylococcus aureus* is both a human commensal and a pathogen, that causes serious nosocomial and community-acquired infections. Despite nostrils being considered its preferred host habitat, the oral cavity has been demonstrated to be an ideal starting point for auto-infection and transmission. The antibiotic resistance assessment of *S. aureus* is a priority and is often reported in clinical settings. This study aimed to explore the prevalence and antimicrobial susceptibility of *S. aureus* in the oral and nasal cavities of healthy individuals. The participants (*n* = 101) were subjected to a demographic and clinical background survey, a caries evaluation, and to oral and nasal swabbing. Swabs were cultured in differential/selective media and *S. aureus* isolates were identified (MALDI-TOF MS) and tested for antibiotic susceptibility (EUCAST/CLSI). Similar *S. aureus* prevalence was found exclusively on nasal (13.9%) or oral (12.0%) habitats, whereas 9.9% of the population were simultaneous nasal and oral carriers. In oro-nasal cavities, similar antibiotic resistance rates (83.3–81.5%), including MDR (20.8–29.6%), were observed. Notably, 60% (6/10) of the simultaneous nasal and oral carriers exhibited different antibiotic resistance profiles between cavities. This study demonstrates the relevance of the oral cavity as an independent colonization site for *S. aureus* and as a potential source of antimicrobial resistance, a role which has been widely neglected so far.

## 1. Introduction

The *Staphylococcus* genus consists of 70 species and subspecies, from which the majority are common colonizers of the human skin and mucous membranes [1]. Although several *Staphylococcus* species are major pathogens in humans, *Staphylococcus aureus* stands out as the most relevant one, considered a High Priority Pathogen by the World Health Organization (WHO) due to causing severe infections worldwide and to the rise and emergence of strains resistant to clinically relevant antibiotics [2]. In particular, methicillin-resistant *Staphylococcus aureus* (MRSA), a term used to describe strains of *S. aureus* that have acquired resistance to β-lactam antibiotics (such as methicillin, amoxicillin, and cephalosporins, except fifth generation), is a major cause of concern, with an estimated 150,000 infections occurring every year in European countries [3,4]. In fact, infections by multidrug-resistant pathogens have become so prevalent that, in 2019, the WHO included antimicrobial resistance as one of the top ten threats to global health [5].

The nostrils are considered the preferred host site of *S. aureus*, with a crucial role in the dissemination of *S. aureus* diseases [6]. *S. aureus* nasal carriages can be divided into persistent, intermittent (or transient), and resistant (non-carriers). Persistent carriers generally carry one single strain at high levels of colonization over time, while intermittent carriers carry varying strains at lower levels of colonization over time [7]. Persistent carriers possess higher counts of *S. aureus* and are as frequent as non-carriers, both having a prevalence of around 30% in the general population, although some studies point to a prevalence of 12–30% in persistent healthy carriers [7,8,9]. Although specific host characteristics, such as the composition of the nasal microbiota, seem to be determinant in *S. aureus* nasal colonization, the mechanisms leading to this process are multifactorial [8]. Vulnerable groups, such as hospitalised, diabetic, obese, allergic, and immunosuppressed patients, have an increased risk for *S. aureus* nasal carriage and, consequently, an increased risk for developing life-threatening infections [8,10,11,12]. According to the Human Oral Microbiome Database (eHOMD) online platform, *S. aureus* colonization is associated with the nasal cavity and the palatine tonsils, (http://www.homd.org, accessed on 1 April 2023), but several other host sites (e.g., skin, rectum, vagina, gastrointestinal and urinary tracts, and oropharynx) can act as reservoirs of this species, playing a role either in the transmission to others or in the transition of this microorganism to other body sites within the host [6,8].

The colonization of the oral cavity by opportunistic/pathogenic microorganisms or the imbalanced oral microbiota are associated with several risks, such as the development of oral diseases (e.g., dental caries, halitosis, periodontitis, oral cancer, systemic infections), and may even impact the progression of multiple systemic diseases (e.g., osteoporosis, atherosclerosis, diabetes, cardiovascular diseases, ischemic cardiomyopathy) [13]. In addition to the alteration from a symbiotic to a dysbiotic microbial community, several other factors are known to have a relevant impact on oral health, such as age, general health, lifestyle, and nutritional status [14]. Particularly regarding the oral carriage of *S. aureus*, evidence is still severely lacking in comparison to nasal colonization. Reported oral carriage rates vary from 24% to 84%, depending on the studied population, yet *S. aureus*’s status as a member of the oral microbiota remains unclear, since it is still mostly regarded as a transient member in this host site [15,16,17]. So far, the *S. aureus* oral carriage has most frequently been associated with specific groups of patients where carriage rates are higher than normal, such as in children, the elderly, terminally ill patients, in individuals with decreased salivary secretion, with carious lesions, removable dentures, fixed prosthetic restorations, periodontitis, rheumatoid arthritis, and with haematological malignancies [15,18]. Nevertheless, some reports have demonstrated the oral cavity to be an important reservoir of *S. aureus*, indicating it to be an ideal starting point for diffusion within the host through external path or through an hematogenous route, via daily dental hygiene routines or during invasive dental procedures [19,20,21]. As *S. aureus* carriage is a risk factor for infection, screening of the nares of vulnerable populations is already carried out for this species, and particularly for MRSA, typically followed by decolonization with mupirocin to prevent infection and/or transmission [21,22,23,24]. The role of the oral cavity as a *S. aureus* reservoir has, however, remained wildly neglected. Oral screening is not yet routinely carried out in clinical settings. Notably, evidence suggests this practice may be useful in the detection of carriers missed by nasal sampling [18,19]. Moreover, the widespread use of prophylactic antibiotics in dentistry has been associated with the emergence of antibiotic resistance in several commensal microorganisms, including staphylococci [25], and MRSA has been previously isolated from the oral cavity, particularly in dentures-wearers and patients with oral infections [26,27]. For these reasons, understanding the role of the oral cavity as a reservoir of colonization and antibiotic resistance becomes a public health necessity.

Accordingly, this study aimed to evaluate the prevalence and the antimicrobial susceptibility profile of *S. aureus* isolates in the oral cavity of a population of healthy adults and compare the results to the nasal cavity, in order to obtain a better understanding of the role of the oral cavity as a *S. aureus* reservoir and as a potential source of antimicrobial resistance.

## 2. Materials and Methods

### 2.1. Study Design and Population

In this study, a total of 101 healthy adults were invited to participate from February to April of 2020. All participants were previously informed about the aim and procedures of the research and gave their informed consent. Each participant filled out a self-report questionnaire to collect demographic and clinical background information (e.g., age, gender, pregnancy history, smoking habits, hormonal contraception history, atopic dermatitis, obesity, systemic diseases, gingivitis, periodontitis, fixed or removable prostheses) (Figure 1). The WHO decayed, missing, and filled teeth (DMFT) index, conducted by a dental professional, was used to assess and quantify the severity of dental caries history in the participants. The exclusion criteria for this study included the inability to give informed consent, pregnancy, recent history of infection, and antibiotic therapy (<3 months).

### 2.2. Oral and Nasal Bacteria Isolation and Identification

The samples were collected aseptically from oral and nasal cavities. Sterile swabs were inserted into both nostrils and gently turned for about 30 s in each. To collect samples from the oral cavity, the swabs were brushed against the inner cheeks, tongue, teeth, and gums for about a minute. The swabs were immediately processed by streaking the surface of selective and differential culture medium Mannitol Salt Agar (MSA, VWR Chemicals BDH, Leuven, Belgium) and incubated at 37 °C for 48 h (Figure 1).

All colonies with distinct appearances (3 to 6 isolates per sample) were selected and re-isolated in MSA (Figure 1). The selected isolates (*n* = 462) were sub-cultured in Brain Heart Infusion (BHI) agar (Biolab Inc., Budapest, Hungary) and incubated at 37 °C for 18–20 h for species identification by Matrix Assisted Laser Desorption/Ionization Matrix-Assisted Laser Desorption Ionization-Time Of Flight Mass Spectrometry (MALDI-TOF MS, Bruker, Mannheim, Germany), according to the manufacturer’s instructions (Figure 1). All isolates were stored at −80 °C in BHI broth (BHI, Biolab Inc., Budapest, Hungary) with 10% glycerol until being used for further studies (Figure 1).

### 2.3. Antibiotic Resistance Assessment

Antibiotic susceptibility was tested in all *S. aureus* isolates by disc diffusion method (Figure 1), following the European Committee of Antimicrobial Susceptibility Testing guidelines (2022) [28] or, when not possible, the Clinical and Laboratory Standards Institute guidelines (2022) [29]. *S. aureus* ATCC^®^ 29213 was used as a quality control strain. The susceptibility to several clinically relevant antibiotics was studied, including those commonly prescribed in general dental practice, namely amoxicillin (10 μg), cefoxitin (30 μg), chloramphenicol (30 μg), ciprofloxacin (5 μg), clindamycin (2 μg), erythromycin (15 μg), gentamicin (10 μg), quinupristin-dalfopristin (15 μg), tetracycline (30 μg), and trimethoprim-sulfamethoxazole (25 μg) (Liofilchem^®^, Roseto degli Abruzzi, Italy). Cefoxitin was used to predict the presence of MRSA strains [28,29]. Multidrug-resistance (MDR) was considered when the isolates were resistant to three or more antibiotics of different classes.

### 2.4. Data Analysis

The results were subjected to statistical analysis (Figure 1) using the Statistical Package for the Social Sciences (IBM^®^ SPSS^®^ Statistics, SPSS Inc., Chicago, IL, USA, 26.0 version). The categorical variables were described through relative frequencies (%) and analyzed by the Chi-square test applying continuity correction, and Fisher’s Exact Test when cells had expected counts less than 5. The non-categorical variable was analyzed by *t*-test and described using mean ± standard deviation (SD). For each test, the statistical significance was set at a α of 5%.

## 3. Results

### 3.1. Population Characterization

The clinical and demographic information of the 101 participants is included in Table 1. The studied population exhibited a mean age of 21.8 ± 3.5 years old, ranging from 18 to 45 years, and was mainly composed of female subjects (82.1%). Overall, for the different factors analyzed, no significant differences were observed. In addition, no cases of periodontitis or removable prosthesis were reported. Regarding systemic diseases, one case of Hashimoto’s thyroiditis, one of psoriasis, and three of chronic gastritis were reported.

### 3.2. Prevalence of S. aureus

A total of 51 *S. aureus* isolates (24 oral and 27 nasal) were observed in 35.6% of the population (*n* = 36 carriers). *S. aureus* prevalence, as depicted in Figure 1, was similar between exclusive oral (12.0%) and nasal (13.9%) carriers (*p* = 0.248). The simultaneous nasal and oral carriers included 9.9% of the participants (Figure 2).

### 3.3. S. aureus Carriage and Its Correlation with Clinical and Demographic Factors

The analysis of *S. aureus* prevalence in the different colonization sites (oral and/or nasal) in relation to different factors, such as gender, is depicted in Table 2. No statistically significant differences (*p* > 0.05) were observed regarding *S. aureus* colonization and gender, hormonal contraception intake, or DMFT index. Additional correlations between *S. aureus* carriage and other specific factors, such as smoking habits, obesity, gingivitis, fixed prostheses, asthma and allergies, and atopic dermatitis, were not carried out due to the very limited number of participants with these conditions.

### 3.4. Antibiotic Resistance

Regarding antibiotic susceptibility, 86.1% of *S. aureus* carriers (31/36) exhibited isolates resistant to at least one antibiotic, while 27.8% (10/36 carriers) exhibited MDR. 

Concerning *S. aureus* isolates, antibiotic resistance was detected in 82.4% (42/51) of the isolates, including MDR at 25.5% (13/51) (Table 3). Similar percentages of resistance (oral—83.3%, nasal—81.5%; *p* = 0.579) and MDR (oral—20.8%, nasal—29.6%; *p* = 0.534) were observed in both oral and nasal isolates (Table 3). Globally, the isolates presented high rates of resistance to several antibiotics, namely 66.7% to gentamicin, 43.1% to amoxicillin, and 31.4% to erythromycin and clindamycin (Table 3). Only 3.9% of the isolates were resistant to tetracycline, and no resistance was detected to the other studied antibiotics, namely, cefoxitin, ciprofloxacin, chloramphenicol, trimethoprim-sulfamethoxazole, and quinupristin-dalfopristin (Table 3). As resistance to cefoxitin was not detected, the absence of MRSA strains was inferred. 

Moreover, a multiplicity of antibiotic resistance phenotypes was detected, as shown in Table 4, but no significant differences were found between the nasal and oral cavities regarding resistance phenotypes (*p* = 0.696). Interestingly, of the 10 participants with oral and nasal simultaneous carriage, 60% (6 carriers) exhibited different antibiotic resistance profiles between cavities.

## 4. Discussion

The results of our study highlight the relevance of the oral cavity as a habitat of equal importance to the nasal cavity regarding *S. aureus* colonization and antibiotic resistance, and alert the healthcare community to the need for the simultaneous screening of nasal and oral *S. aureus* when surveillance and effectiveness of infection control are required.

Regarding the nasal cavity, our results (23.8% nasal carriers) are in accordance with previous works showing similar *S. aureus* carriage rates (21.6–43.8%) among healthy adults, including healthcare workers [30,31,32,33,34]. Around half of *S. aureus* carriers exhibited simultaneous oral and nasal carriage, emphasising the possibility of trafficking of this microorganism between the oral and nasal cavities. The other half (~10% of the studied population) exhibited exclusive oral or nasal *S. aureus* colonization, which suggests and reinforces the possibility raised by some authors of *S. aureus* colonizing the oral cavity without necessarily being present in the nares of the same individual [16,17]. These results demonstrate the importance and the independence of the oral cavity as a reservoir of *S. aureus*, further highlighting the necessity for oral screening in order to detect non-nasal carriers and effectively prevent infection and transmission. Moreover, even in the participants of our study with simultaneous oral and nasal carriage of *S. aureus*, more than half presented different antibiotic resistance profiles between cavities, further reinforcing the possibility of the oral cavity as an independent reservoir of *S. aureus*.

However, it remains controversial whether *S. aureus* belongs to the oral microbiota or if it is just a transient member due to the close anatomic connection with the nasal cavity. For example, in the eHOMD platform, *S. aureus* appears as exclusively of nasal origin, with slight colonization of the palatine tonsils (http://www.homd.org, accessed on 1 April 2023). Thereby, studying the oral and nasal carriage over a large span of time will be necessary to deepen the knowledge on this matter. Nevertheless, as previously mentioned, the prevalence of *S. aureus* in the oral and nasal cavities in this study was similar (21.8% and 23.8%, respectively), which strongly suggests its possible integration in the human oral microbial community, as already raised in previous works [16].

Regarding the correlation between *S. aureus* carriage and clinical and demographic factors, the discrepancies of *S. aureus* carriage rates were not statistically significant in any habitat. Moreover, no statistically significant differences (*p* > 0.05) were observed regarding gender, hormonal contraception intake, or DMFT index (Table 2). As previously mentioned, a correlation between *S. aureus* carriage and other factors (smoking habits, obesity, oral issues, asthma and allergies, and atopic dermatitis) was not carried out due to the very limited number of participants with these conditions. In order to better understand the prevalence of *S. aureus* in this population, it would be interesting to discriminate persistent carriers from intermittent carriers. Hence, the sample collection should be carried out two or more times, at least one week apart, and the evaluation of *S. aureus* should be quantitative, since persistent carries have higher loads of *S. aureus* and are consequently at a greater risk of acquiring *S. aureus* infections [6].

Considering antibiotic resistance, the oral isolates of *S. aureus* exhibited similar resistance rates and profiles to the nasal cavity. The resistance rates (to at least one antibiotic) of the oral isolates exceeded 80% and MDR surpassed 20%, both values being similar to the ones observed in the nasal cavity (81.5% and 29.6%, respectively, *p* > 0.05). In particular, high resistance rates were observed for the several antibiotics (gentamycin, amoxicillin, erythromycin, and clindamycin) (Table 3) most commonly used in dentistry, but also used as alternatives to treat nosocomial staphylococci infections, including methicillin-sensitive *S. aureus* (MSSA) infections [35]. Indeed, these antibiotic resistances have been frequently associated with mechanisms known to be easily acquired by other commensal and/or pathogenic staphylococci by the horizontal gene transfer of mobile genetic elements, therefore favoring their dissemination [36]. Additionally, high resistance and carriage rates of oral *S. aureus* have been previously reported, with carriage rates in the oral cavity varying greatly from study to study, from 24% to values as high as 84%, even in healthy individuals [15,16,17,33,34]. Interestingly, Kearney et al. reported a carriage of oral MSSA of 31.3% in healthcare workers across nine inpatient wards over a two-year period [37]. Compared to the literature, the rate of oral carriage observed in the present study (22%) seems to be on the lower end of the spectrum of what has been reported. Antibiotic resistance in oral *S. aureus* has been observed more frequently in recent years, with MRSA oral carriage being particularly reported [25,26,27,37]. Although MRSA isolates were not detected in this study, resistance rates to other antibiotics are still relevant. A recent study by Garbacz et al. assessed antibiotic resistance in oral staphylococci and obtained high percentages of *S. aureus* resistance to several antibiotics (44% resistance to gentamicin, 35% to tetracycline, 19% to erythromycin, 18% to clindamycin, 12% to cefoxitin, 3% to trimethoprim-sulfamethoxazole, and 1% to ciprofloxacin) and a percentage of MDR of 23% [25]. In this study the isolates also exhibited high rates of resistance to gentamicin (~67%), erythromycin, and clindamycin (~31% each), although the resistance to other antibiotics was lower (Table 3). Furthermore, a similar percentage of MDR isolates, of around 26%, was observed. Considering that this study portrays healthy individuals with no history of concomitant systemic diseases or recurrent antibiotic use, the results are alarming and may even suggest the need for new dentistry therapeutic options, as the use of prophylactic antibiotics in dentistry has been associated with the emergence of antimicrobial resistance in recent years [25,26]. 

Remarkably, multiple resistance profiles were detected in this study (Table 4), with the phenotypes observed among oral isolates being similar to those of nasal isolates, which reinforces that the oral cavity, in parallel with the nasal cavity, is most likely to be an important reservoir of resistant staphylococci. Nevertheless, different profiles between both cavities were observed in the same subjects, suggesting that both cavities are colonized by different *S. aureus* strains, which highlights the role of oral cavity as an independent reservoir of *S. aureus*, including antibiotic-resistant *S. aureus*. Therefore, this study provides evidence that the oral cavity can represent an important reservoir and, consequently, a potential source of resistant bacteria, both to other people through cross-infection and to other regions of the human body by auto-infection. In fact, the oral cavity may be a privileged site of endogenous dissemination within the host, given that daily routines of oral hygiene and invasive dental procedures may promote oral microbial translocation to circulation, potentially causing infection in vulnerable populations [20].

As already mentioned, one drawback of this study is the fact that persistent and intermittent carriers were not distinguished; this distinction would be important in order to further clarify our results regarding nasal and oral carriage. Moreover, further longitudinal studies with larger cohorts of participants would be helpful in assessing how the additional demographic and clinical factors not explored in this study may influence *S. aureus* carriage. It is important to note that participants self-reported their clinical and demographic information, which could introduce bias in the data. In the future, strain-level analysis studies will be crucial (e.g., whole-genome sequencing). This analysis will allow us to reinforce our understanding of the role of the oral cavity as an independent colonization site for *S. aureus*. What is more, it will give relevant information regarding antibiotic resistance genes, and their location (to assess the potential of horizontal gene transfer), virulence factors (to assess the potential of pathogenicity), and clonal linages.

Nonetheless, this study reinforces the idea that the detection of *S. aureus* in the oral cavity, not only in the nasal cavity, is crucial in properly detecting this potential pathogen and avoiding further infection. In fact, exclusive nasal screening is not enough, as demonstrated by the fact that some of our participants were *S. aureus* oral carriers while testing culture negative for nasal carriage. In fact, Donkor and Kotey suggested that oral *S. aureus* colonization could partly explain why decolonization programs targeting only the nasal cavity are often met with failure [26]. As such, we urge the scientific and medical communities to recognize the oral cavity as a relevant staphylococci and antibiotic resistance reservoir, and to include it in screening programs alongside the nasal cavity.

In summary, the present study stresses the relevance of the oral cavity as an independent colonization site for *S. aureus* and as a potential source of antimicrobial resistance to commonly prescribed antibiotics, a role which has been widely neglected so far. Consequently, these findings highlight the need for the oral cavity to be included in surveillance and decolonization programs among healthcare workers and vulnerable patients, in order to prevent the transmission and infection of *S. aureus*.

## Figures and Tables

**Figure 1 pathogens-12-00765-f001:**
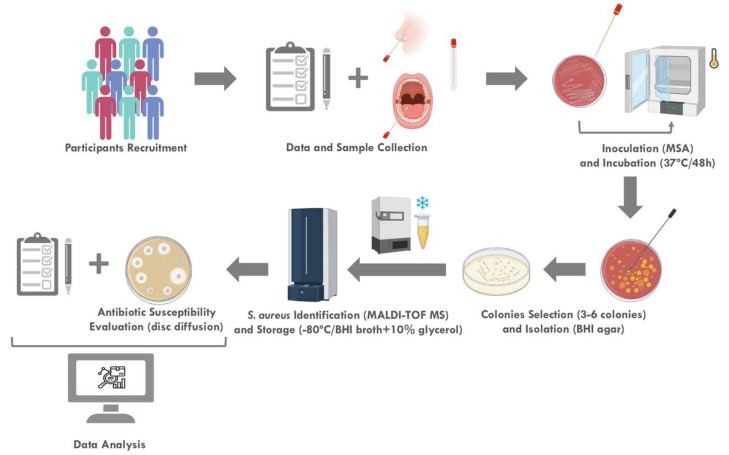
Schematic representation of the methodology used in this study, including participant recruitment, sample collection, and data analysis. MSA, mannitol salt agar; BHI, brain heart infusion; and MALDI-TOF MS, matrix-assisted laser desorption/ionization desorption ionization-time of flight mass spectrometry.

**Figure 2 pathogens-12-00765-f002:**
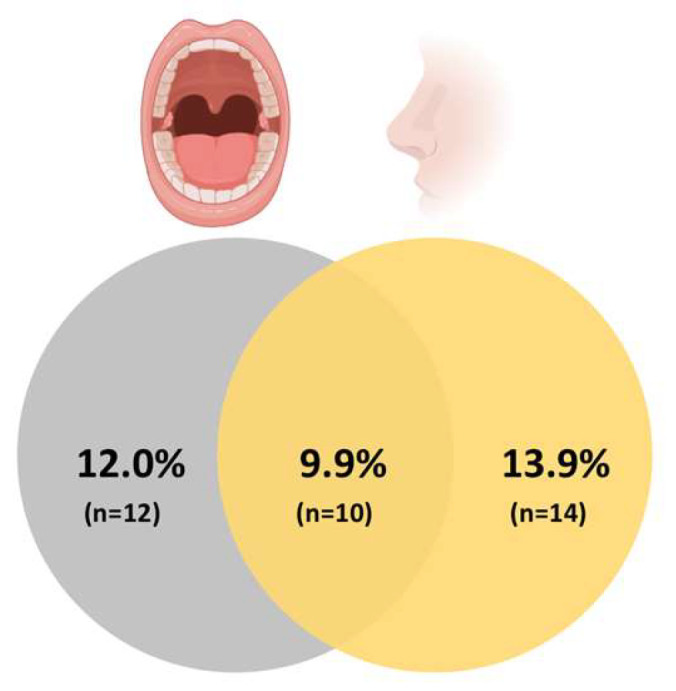
Prevalence of *S. aureus* exclusive oral carriage, exclusive nasal carriage, and oral and nasal simultaneous carriage (overlap). *n*, number of *S. aureus* carriers.

**Table 1 pathogens-12-00765-t001:** Clinical and demographic characterization of the studied population.

Clinical and Demographic Factors	
Sex (female; male)	82.1%; 17.9%
Age (years)	21.81 ± 3.53
Smoking habits	2.29%
Hormonal contraception ^a^	64.19%
Obesity	5.66%
Asthma and allergies	7.02%
Atopic dermatitis	2.67%
Fixed prothesis	3.54%
Gingivitis	4.35%
DMFT	2.60 ± 2.62

Results are shown in prevalence (%) or as mean ± SD. ^a^ Female intake percentage. DMFT, decayed, missing, and filled teeth index.

**Table 2 pathogens-12-00765-t002:** *S. aureus* prevalence in the different colonization sites (oral and/or nasal) in relation to gender, hormonal contraception, and DMFT index.

Clinical and Demographic Factors		Oral Carriers	*p*-Value ^b^	Nasal Carriers	*p*-Value ^b^	Oral and Nasal Carriers	*p*-Value ^b^
Gender	Male	17.6%	>0.999	23.5%	>0.999	11.8%	0.661
	Female	21.8%		23.1%		9.0%	
Hormonal contraception ^a^	Taking	20.0%	0.820	20.0%	0.561	6.0%	0.243
	Not taking	25.0%		28.6%		14.3%	
DMFT	=0 teeth	19.2%	>0.999	19.2%	0.776	3.8%	0.449
	>0 teeth	21.7%		24.6%		11.6%	

DMFT, decayed, missing, and filled teeth index. ^a^ Female intake percentage. ^b^ Chi-square test, applying continuity correction and Fisher’s Exact Test when cells have expected counts less than 5.

**Table 3 pathogens-12-00765-t003:** Antibiotic resistance rates of the *S. aureus* isolates according to origin (oral or nasal).

	Total	Oral	Nasal
	(51)	(24)	(27)
R	82.4% (42)	83.3% (20)	81.5% (22)
MDR	25.5% (13)	20.8% (5)	29.6% (8)
AML	43.1% (22)	45.8% (11)	40.7% (11)
FOX	0% (0)	0% (0)	0% (0)
CIP	0% (0)	0% (0)	0% (0)
C	0% (0)	0% (0)	0% (0)
CN	66.7% (34)	58.3% (14)	74.1% (20)
TE	3.9% (2)	4.2% (1)	3.7% (1)
SXT	0% (0)	0% (0)	0% (0)
E	31.4% (16)	29.2% (7)	33.3% (9)
CD	31.4 % (16)	29.2% (7)	33.3% (9)
QDA	0% (0)	0% (0)	0% (0)

R, Resistance (isolates resistant to at least one antibiotic); MDR, multidrug-resistance (isolates resistant to three or more antibiotics of different classes); AML, amoxicillin (used as a representative of penicillinase-labile penicillins); C, chloramphenicol; CD, clindamycin; CIP, ciprofloxacin; CN, gentamicin; E, erythromycin; FOX, cefoxitin; QDA, quinupristin-dalfopristin; SXT, trimethoprim + sulfamethoxazole; TE, tetracycline; and (*n*), number of isolates.

**Table 4 pathogens-12-00765-t004:** The different phenotypic resistance profiles detected of the *S. aureus* isolates.

Phenotypic Resistance Profiles	Total	Oral	Nasal
	(51)	(24)	(27)
-	17.6% (9)	16.7% (4)	18.5% (5)
AML	7.8% (4)	12.5% (3)	3.7% (1)
CN	17.6% (9)	12.5% (3)	22.2% (6)
AML-CN	23.5% (12)	25% (6)	22.2% (6)
CD-E	5.9% (3)	8.3% (2)	3.7% (1)
CN-TE	2.0% (1)	4.2% (1)	0% (0)
AML-CD-E	2.0% (1)	4.2% (1)	0% (0)
CD-CN-E	13.7% (7)	12.5% (3)	14.8% (4)
AML-CD-CN-E	7.8% (4)	4.2% (1)	11.1% (3)
AML-CD-CN-E-TE	2.0% (1)	0% (0)	3.7% (1)

-, isolates susceptible to all antibiotics tested; AML, amoxicillin (used as a representative of penicillinase-labile penicillins); CD, clindamycin; CN, gentamicin; E, erythromycin; TE, tetracycline; and (*n*), number of isolates.

## Data Availability

The data presented in this study are available on request from the corresponding author. The data are not publicly available due to privacy reasons.

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
