# Peer review of "Unveiling the Relevance of the Oral Cavity as a Staphylococcus aureus Colonization Site and Potential Source of Antimicrobial Resistance"

_pathogens, 2023, doi:10.3390/pathogens12060765_

Round 1

Reviewer 1 Report

This study mainly discussed the relevance of the oral cavity as a habitat of equal importance to the nasal cavity. And a total of 101 healthy adults is involved in this study.

In introduction, “S. aureus nasal carriage can be divided into persistent, intermittent (or transient), and resistant (non-carriers).”. By which standard the above type was defined. This standard is hoping to be explained.

S. aureus should in an Italic type such as “Antibiotic susceptibility was tested in all S. aureus isolates”

I believe a brief scheme about how S. aureus was identified, combined with your description will make your method section clearer.

Is there any reference for authors to determine the items of Clinical and demographic factors?

In your results, “Clinical and demographic factors” and S. aureus isolates were discovered. How those Clinical and demographic factors effect on S. aures is not mentioned in discussion. If possible, please added your rational supposition with associated references

In your discussion, the drawbacks and improvement in the future of your study should be mentioned.

The English language is well with moderate grammar errors

Author Response

In reply to the reviewers performed on the paper entitled “Unveiling the relevance of the oral cavity as a Staphylococcus aureus colonization site and potential source of antimicrobial resistance”, we would like to acknowledge the valuable comments performed by the editor that kindly accepted to revise our manuscript. We would like to confirm that we have addressed most issues and answered the questions made by the two reviewers. We hope the answers below and modifications that have been done in the manuscript are clear and concise enough as required by the reviewer to enable the publication of the manuscript in Pathogens.

Answer to referee’s comments and queries

Detailed responses to Reviewer 1

 Reviewer´s comment (1) - In introduction, “S. aureus nasal carriage can be divided into persistent, intermittent (or transient), and resistant (non-carriers).”. By which standard the above type was defined. This standard is hoping to be explained.

Our reply: We thank the reviewer for the question. We have added the following clarification regarding the standard (lines 47-49): “Persistent carriers generally carry one single strain at high levels of colonization over time, while intermittent carriers carry varying strains at lower levels of colonization over time [8].”

Reviewer´s comment (2) - S. aureus should in an Italic type such as “Antibiotic susceptibility was tested in all S. aureus isolates”

Our reply: We thank the reviewer for the suggestion. The correction was carried out in the 2.3. Antibiotic resistance assessment subsection.

Reviewer´s comment (3) - I believe a brief scheme about how S. aureus was identified, combined with your description will make your method section clearer.

Our reply: We thank the reviewer for the suggestion. A schematic figure, Figure 1, was added to better clarify the methods section.

Reviewer´s comment (4) - Is there any reference for authors to determine the items of Clinical and demographic factors?

Our reply: We thank the reviewer for the question. We thank the reviewer for the question. Indeed, the authors based the choice of these items in previous evidence reported to be associated with S. aureus colonization, namely: sex (PMID: 26202769); age (PMID: 32174894); Hormonal contraception (PMID: 22955426); Smoking (PMID: 35269394, PMID: 34589469, PMID: 29311241); Obesity (PMID: 23667661); dental prothesis (PMID: 15800465); and allergy (PMID: 28679656, PMID: 32460514). Also, oral health status of our population was partially assessed by gingivitis prevalence and DMFT index.

Reviewer´s comment (5) - In your results, “Clinical and demographic factors” and S. aureus isolates were discovered. How those Clinical and demographic factors effect on S. aureus is not mentioned in discussion. If possible, please added your rational supposition with associated references

Our reply: We thank the reviewer for the suggestion. We have added the following paragraph to the discussion (lines 246-250): “Moreover, no statistically significant differences (p>0.05) were observed regarding gender, hormonal contraception intake, or DMFT index (Table 2). As previously mentioned, correlation between S. aureus carriage and other factors (smoking habits, obesity, oral issues, asthma and allergies and atopic dermatitis) was not carried out due to the very limited number of participants with these conditions.”

Reviewer´s comment (6) - In your discussion, the drawbacks and improvement in the future of your study should be mentioned.

Our reply: We thank the reviewer for the suggestion. We have added the following paragraph to the discussion (lines 300-311): “As already mentioned, one drawback of this study is the fact that persistent and intermittent carriers were not distinguished; this distinction would be important in order to further clarify our results regarding nasal and oral carriage. Moreover, further longitudinal studies with larger cohorts of participants would be helpful in assessing how additional demographic and clinical factors not explored in this study may influence S. aureus carriage. It is important to note that participants self-reported their clinical and demographic information, which could introduce bias in the data. In the future it will be crucial the strain-level analysis studies (e.g., whole-genome sequencing). This analysis will allow to reinforce and understand the role of the oral cavity as an independent colonization site for S. aureus. Also, it will give relevant information regarding antibiotic resistance genes, and their location (to assess the potential of horizontal gene transfer) virulence factors (to assess the potential of pathogenicity), and clonal linages.”

Reviewer 2 Report

This study edited by Campos et al. investigated the prevalence of S. aureus in healthy adults and fond that similar prevalence was detected on both nasal and oral cavities. Furthermore, they evaluated their antibiotic resistance pattern. Consequently, the authors concluded that the oral cavity is an important reservoir of S. aureus and potential source of antimicrobial resistance. Overall, this study provides interesting and valuable information to the readers, especially researchers and stuffs in the public health area. But there are some unclear descriptions which need to be edited.

Specific comments are as follows:

1. Result section 3.2:

In Fig. 1, the authors present only the percentage data (12.0%, 9.9%, and 13.9%). The number of adults who are S. aureus-positive should be presented for clarity, this reviewer calculated from the data to be 12.0% (12 adults), 9.9% (10 adults), and 13.9% (14 adults). If the calculation is correct, the description in the abstract “55% of the population … (lines 25–26)” seems incomprehensible. 

In addition, the numbers of isolates from oral cavity (24 isolates) and nasal cavity (27 isolates) were presented, but information of isolates from the same adult was missing. Based on the data, this reviewer cannot understand how many S. aureus isolates are obtained from the same adults.

2. Table 3:

Tetracycline resistance (3.9%) should be included in Table 3. In addition, the number of isolates for each antibiotic resistance profile should be presented in addition to % data. From the % data, the sentence “Globally, the isolates… (lines 182–184)” is confusing. For example, gentamicin resistance can be calculated to be 68%, not 66.7%, from the data of Table 3. 

3. Lines 190–191:

The sentence “55% of the participants…” is not clear. What the value “55%” stand for? This reviewer understands that 10 participants are positive both on oral and nasal cavities. Thus, the explanation is confusing due to lack of precise presentation of the number of adults                                                        in the previous section 3.2.

4. The terms “microbiome” and “microbiota” used in this manuscript are confusing. “Microbiome” is a term describing genes of microorganisms, in this study “microbiota”. For example, the term “microbiome” in line 207 is “microbiota”. Please check them.

5. Lines 205–207: If the statement of the sentence “Around half of…” is correct, one would expect that the antibiotic resistance pattern of S. aureus isolated form oral is the same as that of isolates from nasal in the same person. But, the authors explained the results in another way in lines 266–269. How do the authors explain this discrepancy?

Author Response

Dear Editor of Pathogens,

In reply to the reviewers performed on the paper entitled “Unveiling the relevance of the oral cavity as a Staphylococcus aureus colonization site and potential source of antimicrobial resistance”, we would like to acknowledge the valuable comments performed by the editor that kindly accepted to revise our manuscript. We would like to confirm that we have addressed most issues and answered the questions made by the two reviewers. We hope the answers below and modifications that have been done in the manuscript are clear and concise enough as required by the reviewer to enable the publication of the manuscript in Pathogens.

Detailed responses to Reviewer 2

Reviewer´s comment (1) - Result section 3.2:

In Fig. 1, the authors present only the percentage data (12.0%, 9.9%, and 13.9%). The number of adults who are S. aureus-positive should be presented for clarity, this reviewer calculated from the data to be 12.0% (12 adults), 9.9% (10 adults), and 13.9% (14 adults). If the calculation is correct, the description in the abstract “55% of the population … (lines 25–26)” seems incomprehensible.

Our reply: We thank the reviewer for the comment. As suggested, the number of participants was added to the figure (now Figure 2).

Concerning the “55% of the population”, which was corrected to 60%, this percentage refers to the fact that 60% of the participants, with S. aureus isolates recovered from both cavities (6 out of 10 participants), exhibited different antibiotic resistance profiles between cavities (different profiles between both cavities were observed in the same subjects). This was corrected and clarified in both section 3.5 (lines 204-206) and the abstract (lines 25-26). This result shows that both cavities are colonized by different S. aureus strains, highlighting the role of oral cavity as an independent reservoir of S. aureus (lines 290-293 of Discussion section).

Reviewer´s comment (2) - Table 3: Tetracycline resistance (3.9%) should be included in Table 3. In addition, the number of isolates for each antibiotic resistance profile should be presented in addition to % data. From the % data, the sentence “Globally, the isolates… (lines 182–184)” is confusing. For example, gentamicin resistance can be calculated to be 68%, not 66.7%, from the data of Table 3.

Our reply: We thank the reviewer for the suggestion. Table 3 was completed with additional information regarding antibiotic resistance rates, including numbers of isolates, to clarify the percentages.

Reviewer´s comment (3) - Lines 190–191: The sentence “55% of the participants…” is not clear. What the value “55%” stand for? This reviewer understands that 10 participants are positive both on oral and nasal cavities. Thus, the explanation is confusing due to lack of precise presentation of the number of adults in the previous section 3.2.

Our reply: We thank the reviewer for the comment. As explained above, the “55% of the population”, which was corrected to 60%, refers to the fact that 60% of the participants with S. aureus isolates recovered from both cavities (6 out of 10 participants) exhibited different antibiotic resistance profiles between cavities. This was corrected and clarified in both section 3.5 (lines 204-206) and the abstract (lines 25-26).

Reviewer´s comment (4) - The terms “microbiome” and “microbiota” used in this manuscript are confusing. “Microbiome” is a term describing genes of microorganisms, in this study “microbiota”. For example, the term “microbiome” in line 207 is “microbiota”. Please check them.

Our reply: We thank the reviewer for the suggestion. The term was corrected across the text.

Reviewer´s comment (5) - Lines 205–207: If the statement of the sentence “Around half of…” is correct, one would expect that the antibiotic resistance pattern of S. aureus isolated form oral is the same as that of isolates from nasal in the same person. But, the authors explained the results in another way in lines 266–269. How do the authors explain this discrepancy?

Our reply: We thank the reviewer for the question. When we mention “around half of” in lines 205-207, we are not referring to antibiotic resistance, but to S. aureus carriage. “Around half of S. aureus carriers exhibited simultaneous oral and nasal carriage”. As the total number of carriers (nasal and/or oral) in this study is 26, almost half of these carriers (10) exhibit simultaneous oral and nasal carriage. In lines 266-269, we are discussing percentages of antibiotic resistance, which have different values. We hope that this clarifies the question.
